# Innovating with Nature: From Nature-Based Solutions to Nature-Based Enterprises

Esmee D. Kooijman [1,*], Siobhan McQuaid [2], Mary-Lee Rhodes [2], Marcus J. Collier [3] and Francesco Pilla [1]

1   School of Architecture, Planning and Environmental Policy, University College Dublin, Dublin 4, Dublin, Ireland; francesco.pilla@ucd.ie
2   Trinity Business School, Trinity College Dublin, Dublin 2, Dublin, Ireland; siobhan.mcquaid@tcd.ie (S.M.); rhodesml@tcd.ie (M.-L.R.)
3   School of Natural Sciences, Trinity College Dublin, Dublin 2, Dublin, Ireland; marcus.collier@tcd.ie
*   Correspondence: esmeekooijman@gmail.com

**Abstract:** Nature-based solutions (NBS) to address societal challenges have been widely recognised and adopted by governments in climate change and biodiversity strategies. Nevertheless, significant barriers exist for the necessary large-scale implementation of NBS and market development is still in its infancy. This study presents findings from a systematic review of literature and a survey on private sector agents in the planning and implementation of NBS, with the aim to identify them. In this study, we propose a typology for organisations delivering NBS and a categorisation of their economic activities. The most common organisation type found is nature-based enterprise which offers products or services where nature is a core element and used sustainably and engages in economic activity. Moreover, eleven categories of economic activities were identified, ranging from ecosystem restoration, living green roofs, and eco-tourism to smart technologies and community engagement for NBS. Nature-based enterprises contribute to a diverse range of sustainable economic activities, that standard industry classification systems do not adequately account for. The recognition of the value created by these activities is essential for designing effective policy support measures, and for market development of the sector and its potential to facilitate the wider adoption of NBS.

**Keywords:** nature-based enterprise; nature-based solutions; sustainable development; classification of economic activities

## 1. Introduction

Natural ecosystems provide services of crucial importance to human well-being, health and livelihoods by sustaining the quality of air, water and soils, providing resources and energy, regulating the climate, and reducing the impact of natural hazards [1]. Yet, human drivers as a consequence of global change have significantly altered terrestrial, freshwater and marine ecosystems, and biodiversity loss ranks among the most urgent issues we are facing today [2]. Although the value of ecosystems is difficult to quantify in monetary terms, estimated losses of their services were between $4.3 and $20.2 trillion per year from 1997 to 2011, mainly as a result of global land-use change [3]. Restoring, effectively managing and creating natural ecosystems do not only have the potential to improve ecosystem functions and biodiversity [4] but also to decrease the vulnerability of climate change effects by increasing resilience for adaptation and mitigating greenhouse gas emissions [5]. Stewardship of terrestrial ecosystems and improvement of agricultural methods have the potential to provide up to 30% of the greenhouse gas mitigation required until 2030 to keep global warming to less than 2 °C compared to pre-industrial levels [6].

Nature-based solutions (NBS) cover different approaches that work with and enhance nature to help address societal challenges, including climate change and biodiversity loss. In doing so, NBS generate a wide range of benefits locally and for society as a

whole [7]. Examples of NBS include ecosystem-based adaptation and mitigation, eco-disaster risk reduction, green/blue infrastructure [8], and natural climate solutions [4]. The benefits of implementing NBS to solve environmental challenges—as opposed to traditional approaches—have led to the adoption of the concept by policymakers, however, not universally and to varying degrees of success [9]. In the EU, NBS have been part of the research and innovation funding agenda since 2014, aligning the strategy for biodiversity and ecosystem services with goals of innovation for growth and job creation [10]. More recently, the critical role of NBS was acknowledged for creating a net-zero society by 2050 in the EU Green Deal [11], as well as for climate change adaptation and mitigation in the widely supported NBS manifesto from the UN Climate Action Summit [12].

Nevertheless, implementation of NBS on the scale needed to contribute to these societal challenges requires the involvement of all stakeholders, including the private sector [13,14]. Even though several frameworks and standards for NBS have been published [7,15–17], guidelines for the effective implementation of NBS for practitioners and decision-makers are lacking [18]. Other barriers for NBS implementation include uncertainty about the value delivered by NBS, including economic value, lack of financial resources and expertise, and land and time availability [19]. These barriers mainly relate to the public sector as NBS have mainly been implemented by this sector. Private sector involvement could contribute to overcoming some barriers, and NBS are increasingly viewed as a means to diversify and transform business for sustainable development [6,20].

However, what kind of organisations contribute to the delivery of NBS? And what kind of activities do they undertake? As the market development is still in its infancy, industry classifications of sectors of economic and financial activity do not account for NBS related activities [21]. We address this knowledge gap by exploring the characteristics and activities of organisations supporting the delivery of NBS. More specifically, this research attempts to define these types of organisations by proposing a typology and by providing an overview of the categories of economic activity they are engaged in. It explicitly focuses on Nature-Based Enterprises (NBE), which we define as 'an enterprise, engaged in economic activity, that uses nature sustainably as a core element of their product/service offering'. Here, nature may be used directly by growing, harnessing, harvesting or sustainably restoring natural ecosystems, and/or indirectly by contributing to the planning, delivery or stewardship of nature-based solutions. Researchers and practitioners iteratively developed this definition in the Horizon 2020 Connecting Nature project (www.connectingnature.eu) over the past three years, while taking into account the EC definitions for enterprise [22] and nature-based solutions [23].

This paper is structured as follows: first, we present the foundations for the concept of nature-based enterprise followed by the research methodology. Second, we discuss the research findings and present an organisation typology and overview of economic activities. Third, we address the study's limitations and propose future research directions.

## 2. Theoretical Background

### 2.1. Enterprise and Economic Activity

While entrepreneurship and enterprise are often used interchangeably in academic literature, for this study, we make a distinction. Entrepreneurship is defined as the process of opportunity discovery, creation, and exploitation. Enterprises are the outcome of the process of entrepreneurship: business organisations [24]. The European Commission defines an enterprise as 'any entity engaged in economic activity, irrespective of its legal form' in which economic activity is defined as 'the sale of products or services at a given price, on a given market' [22] (p. 9).

Economic activities are organised into sectors - areas of business that make up a country's economy. One of the classifications of economic sectors is the Statistical Classification of Economic Activities in the European Community (NACE). This reference framework aims to provide statistics on economic activities that are comparable at the European and world level. In NACE, activities are grouped based on common processes for producing

goods or services using similar technologies. It provides market insights for specific sectors irrespective of the type of legal organisation or mode of operation, and thus includes enterprises, individual proprietors and governments [25]. However, NACE is inconsistent and incomplete with regard to the EU's environmental objectives, and important economic activities that contribute to the transition to climate neutral economies are not captured by the NACE codes [21]. Similarly, for ecological restoration activities in the USA no data are collected in standard public data sources and activities are not included in the North American Industry Classification [26].

Nevertheless, entrepreneurship is increasingly recognised as important for bringing innovation and transformation towards sustainable products and processes [27] and contributing to sustainability transitions [28]. Sustainable enterprises aim to solve societal and environmental problems through business activities. As such, they focus on creating sustainable value, i.e., economic, environmental and social value [29]. Several scholars have defined different types of enterprises that contribute to sustainable development. Eco-enterprises [30] and green enterprises [31], contribute to solving environmental problems indirectly, for example, by the creation of sustainable products and processes. Ecological and environmental enterprises contribute to solving environmental problems directly, for example, by restoring nature and biodiversity or decreasing environmental pollution and ecological degradation [32,33]. Nature entrepreneurship is based on resources and experiences offered by nature, and activities of these types of enterprises are characterised by being nature-oriented, responsible, indigenous, local, and handicraft. Examples include nature tourism, harvesting of food products and recreational services [34].

The characteristics of enterprises or ventures can be described through internal organisation, structure, strategy elements (i.e., mission, goals, and impact), and governance [35]. When considering the mission of environmental enterprise, the goal of making profit is usually subordinate and a means to solve environmental problems [29]. In addition, social enterprises have a dual goal of achieving financial sustainability and creating value for society [36]. This dual mission results in a spectrum of three enterprise types, with hybrid enterprises between traditional non-profit and for-profit enterprises, defined by their purpose to create social value, economic value, or both [36]. Although the concept of dual mission is predominantly applied in social enterprise research, this hybridity appears relevant to enterprise types that create both economic and environmental value.

### 2.2. Nature-Based Activities

To connect the economy with nature, the term 'natural capital' is frequently used, in which capital refers to a stock that yields a flow of ecosystem services over time [37]. Ecosystem services are the ecological characteristics, functions or processes that benefit human well-being. They are generally classified into four classes, of which benefits are either received directly through provisioning (nature provides a resource), regulating (nature regulates environmental impacts) or cultural (nature offers social and cultural value) services, or indirectly in case of supporting services. Here, nature supports essential ecosystem functions by maintaining the processes and functions necessary for direct ecosystem services [1,37]. To realise the benefits from ecosystem services, interaction with other forms of capital requiring human agency—built or manufactured capital, human capital, and social or cultural capital—is needed [38].

Nature-based solutions (NBS) is an umbrella concept that covers a range of ecosystem service-related approaches [7]. Encompassing a vast array of interventions, NBS are based on the premise that healthy natural and managed ecosystems produce essential services, e.g., from storing carbon, controlling floods and stabilising slopes to providing clean air and water, food and medicine [6,37]. Nature-based solutions are defined by IUCN as 'actions to protect, sustainably manage, and restore natural or modified ecosystems, that address societal challenges effectively and adaptively, simultaneously providing human well-being and biodiversity benefits' [7] (p. 2). The EC define nature-based solutions as 'inspired and supported by nature, which are cost-effective, simultaneously provide environmental,

social and economic benefits and help build resilience' [23]. Three main types of NBS can be characterised based on the maximisation of the delivery of key services and the level of engineering of ecosystems [15]:

- Type 1 promotes better use of natural/protected ecosystems by no or minimal intervention to maintain or boost the effects of ecosystem services (e.g., mangrove restoration to protect coastlines).
- Type 2 focuses on the effective management towards the sustainability and multifunctionality of ecosystems and landscapes to support selected ecosystem services (e.g., managing naturally occurring parks or forests).
- Type 3 focuses on managing ecosystems through interventions or even creating new ecosystems (e.g., green walls or roofs, newly created urban parks or green infrastructure).

The implementation of NBS, by definition, must benefit biodiversity, support the delivery of a range of ecosystem services, as well as contribute to societal goals [23]. Subsequently, organisations delivering NBS positively contribute to biodiversity and ecosystem services. However, knowledge gaps exist regarding these types of organisations and their economic activities. This lack of clear designation poses a problem for the market development of organisations delivering NBS and their products and services, as we simply do not know the economic value they generate. Moreover, the variety of interventions and types of NBS result in multiple benefits provided that might be valued differently by specific stakeholder groups. This can lead to trade-offs between ecosystem services [6]. Natural forests, grasslands, and wetlands may, for instance, store more carbon or provide higher biodiversity than managed ecosystems such as parks or green roofs. Yet, these managed systems could increase urban cooling, water retention capacity and contribute to overall health and well-being. Furthermore, if the broader ecological context is not considered in the policy or management, this could lead to negative biodiversity and environmental outcomes, or so-called 'bio-perverse' outcomes [39]. NBS interventions could also lead to ecosystem disservices−negatively perceived ecosystem services−for human well-being. An example of this is an increase in allergic pollen as a result of newly created green areas [40].

## 3. Methodology

Following the research objectives and the lack of knowledge in this field, the research design follows a mixed-method approach. This approach is found useful to broaden studies and gain more profound insights by combining both qualitative and quantitative research methods, and by triangulation of results [41]. This study combines data collected by a systematic literature review and an enterprise survey, ensuring it captures both the state-of-the-art academic literature and the empirical evidence the field. A summary of the data collection, validation and analysis process is shown in Figure 1.

### 3.1. Data Collection

3.1.1. Systematic Literature Review

The systematic literature review was conducted in SCOPUS between February and June 2020 and followed the Preferred Reporting Items for Systematic Reviews (PRISMA, www.prisma-statement.org) guidelines. SCOPUS was selected for its broad coverage compared to other academic search engines. The search combines the two underlying concepts of nature-based activities and enterprise. Keywords were constructed based on results from scoping searches, academic thesauri, expert feedback and relevant systematic reviews [19,33,42–44]. Exclusion criteria were applied in the search string to increase the relevance of the results. There were no restrictions imposed on the type of publication to capture all existing relevant studies. We only included articles written in the English language. Table 1 shows the final search query.

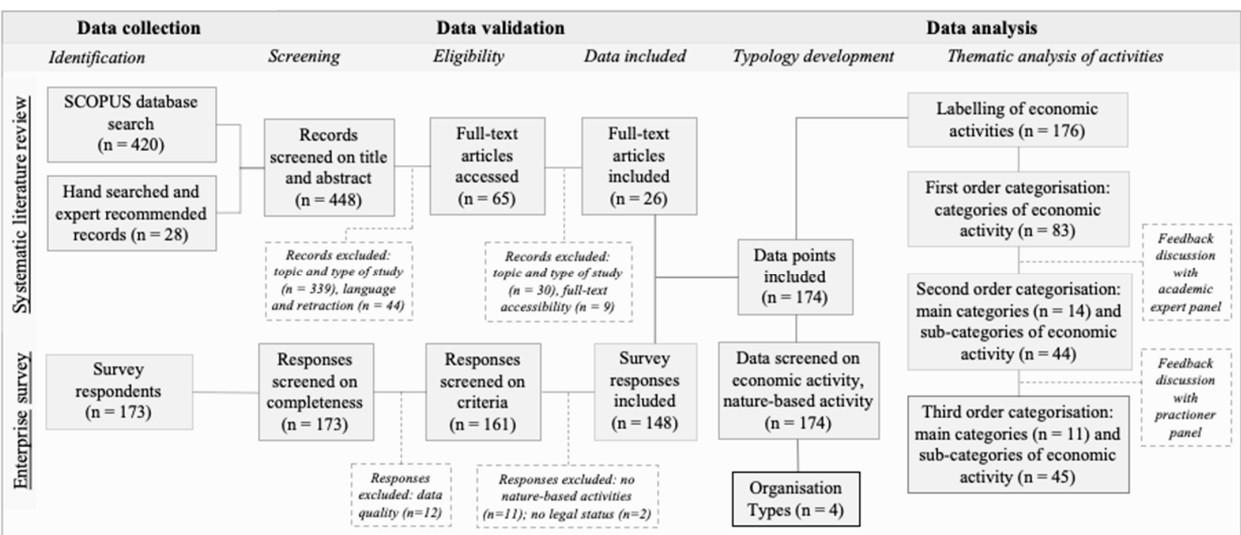

**Figure 1.** Overview of the data collection, validation and analysis process.

**Table 1.** Search string used for systematic literature review in SCOPUS.

| Category | Search String |
|---|---|
| Nature-based solutions | KEY ("nature-based solution" OR "nature-based" OR "ecosystem-based" OR "ecosystem service" OR "climate change adaptation" OR "climate change mitigation" OR "biodivers*" OR "nature" OR "green" OR "ecolog*" OR "blue infrastructure" OR "sustainable urban drainage system" OR "constructed wetland" OR "green space" OR "green infrastructure" OR "urban park" OR "forest*" OR "green building" OR "green roof" OR "green wall" OR "community garden" OR "urban farm" |
| Enterprise | W/3 "enterprise" OR "entrepreneur*" OR "business" OR "business organisation" OR "venture" OR "social enterprise") |
| Exclusion criteria [1] | AND NOT ABS (springer AND business OR nature) AND NOT ABS ("nature of") AND PUBYEAR > 2009 |

[1] For example, we excluded records of which the abstract only contained the words 'nature of' or 'springer' in combination with 'nature' or 'business', as this respectively resulted in records that discussed the nature of anything (e.g., nature of business) and that were published in Springer Nature or Business journals. Moreover, as the term nature-based solutions only started to be used in the late 2000s [45], records from before 2010 are excluded.

### 3.1.2. Enterprise Survey

The enterprise survey was publicly available from February to June 2020 in 13 languages and was distributed through the networks of Connecting Nature cities. The systematic literature review informed the survey questions, and the survey was tested and adapted after several feedback rounds with six SMEs. The survey consisted of 44 questions in 4 sections: question 1–16 focused on the general enterprise characteristics, question 17–32 on activities, question 33–40 on value creation and question 41–44 on barriers and enablers experienced. In this paper, only data collected in the first two sections are discussed. Most survey questions were structured questions, and open-ended questions explicitly focused on the enterprise's activities, services and product offerings to capture the potential breadth and diversity.

### 3.2. Data Validation

The data from the literature review and the survey were assessed on the criteria of nature-based activities and enterprise (Section 2). To summarise, the majority of activities are nature-based, i.e., directly or indirectly related to the delivery of NBS, and executed by an organisation that is an enterprise, i.e., a legal form of organisation that engages in economic activity. From the literature review, 26 papers were included based on these criteria. The search resulted in 420 records and 28 more were included by hand searching and expert recommendations. The 448 records were screened using eligibility criteria, after

which 26 relevant were included in the study. Records dealing solely with environmental aspects or enterprise/business aspects and records on education, research or policy—as opposed to entrepreneurial activities—were excluded. Grey literature was considered [46–48], but their focus did not fit the scope and were not included in the review.

Of the 173 survey responses, 148 were included. The reasons for exclusion of responses were low data quality and completeness (12), lack of legal status (2) or the organisation's activities were not nature-based (11). In the case of the latter, these organisations were economic and tourism development organisations (3), renewable energy companies (3), sustainable mobility companies (2) or food producers (2) and education institutions (1) without a focus on sustainability and/or NBS. The survey relied primarily on self-reporting by the respondents; however, in case of incomplete or contradictory data, a web search was conducted to validate data and supplement gaps.

### 3.3. Data Analysis

As a result of the data validation process, a total of 174 publications and survey responses were uploaded to NVivo12, a software tool designed to facilitate qualitative and quantitative data analysis.

### 3.3.1. Typology Development

The overarching concept in this study are the delivery of NBS by different types of organisations, with the main aim to define the private sector organisations. To describe the characteristics of these type of organisations, a typology was developed. Typologies are organised systems of types that contribute to a variety of analytic work, including forming and refining concepts, creating categories for classification and measurement and sorting cases [49]. They are widely used in strategic management and organisational literature, mainly because this form of theory-building can describe causal relationships of contextual, structural, and strategic factors [50]. As classifications grouped according to general types, one of the strengths of typologies is that they take into account multiple elements in a complex analysis that are resulting in a relatively simple end product [51]. Typologies are based on an overarching concept disaggregated into two or more dimensions. The categories of these dimensions establish the rows and columns in the typology matrix, of which the cells contain the resulting concepts, or different types in the typology [49]. Our approach to develop an organisation typology for these types of organisations is therefore based on the two criteria for nature-based enterprise, as defined in Section 2: (1) engagement in economic activity, i.e., sell products or services for a given price on a market, and (2) direct or indirect contribution to the delivery of nature-based solutions, and thereby positively contribute to biodiversity and ecosystem services. We applied these criteria to the 174 data points from the survey and literature review to develop the typology presented in Section 4.1.

### 3.3.2. Thematic Analysis of Activities

Activities from literature related to the direct and indirect use of nature were labelled based on the theme. The same was done for the open-ended questions on activity, product and service offerings from the survey responses. All activities were coded, after which NVivo12 allowed us to retrieve all text segments with a particular label and to read those segments alongside each other. To analyse the data, a grounded theory approach was used. Grounded theory generates knowledge from the data collected in one or more empirical studies, as opposed to existing theory [52]. Based on this iterative process, we organised the data into key themes [53]. Before finalising, the themes were discussed with a panel of academic experts and practitioners in separate feedback sessions. This resulted in 11 main categories of economic activity and 45 sub-categories, split over direct and indirect use of nature, as presented in Section 4.2. The final categories of economic activity were assigned to the closest NACE classification by comparing to different levels of the NACE classification and the corresponding descriptions.

## 4. Findings

The systematic literature resulted in the selection of 26 publications: 10 studies are empirical and mainly use small samples of quantitative data and case studies, 5 are review papers and 11 use a mixed-method approach, i.e., quantitative and qualitative methods. Studies covered European countries (13), the Americas (4), China (2) and the Arctic (1). Of the 148 validated survey responses, 35 organisations were established before the year 2000, 36 between 2000 and 2009 and 77 between 2010 and 2019. Most organisations were from Europe (90%) and operate either on a national, regional or local scale (91%). The findings from literature and the survey are presented together in this section.

### 4.1. Types of Organisations Delivering NBS

Nature-based solutions are delivered by different types of organisations, including enterprises. Section 2 provides the framework for the criteria for nature-based enterprise used for the evaluation of the 174 data points in this research, namely: (1) engagement in economic activity, i.e., sell products or services for a given price on a market, and (2) direct or indirect contribution to the delivery of nature-based solutions, and thereby positively contribute to biodiversity and ecosystem services. We found that not all organisations are active on economic markets or have activities exclusively related to nature-based solutions. Around two-thirds (68%) of the survey respondents consider themselves to be a nature-based enterprise, and of the organisations found in literature not all their activities–mainly in forestry (12), tourism (7), nature conservation and restoration (4), agriculture and food production (3), smart green solutions (2) and green care (1)—would qualify under the criterium 'nature-based'

However, organisations that do not fit within the criteria of nature-based enterprise could contribute to the delivery of NBS. For example, governmental organisations, third organisations (e.g., community groups, associations, charities etc.) do deliver NBS without selling products or services, and other business organisations as well, even though it might be part of a wider portfolio of activities. In order to capture all relevant types of organisations delivering NBS, we propose a typology based on the parameters (1) engagement in economic activity and (2) the use of nature (Table 2).

**Table 2.** Types of organisations delivering nature-based solutions.

|  | Nature is at the Core of Activities | Nature is Not at the Core of Activities |
|---|---|---|
| **Economic activity** | Nature-based enterprise | Enterprises delivering nature-based products and services |
| **No economic activity** | Nature-based organisation | Organisations delivering nature-based products and services |

- Nature-based enterprises use nature as a core element of their product/service offering for the planning, delivery and/or stewardship of NBS and engage in economic activity.
- Nature-based organisations use nature as a core element of their product/service offering for the planning, delivery and/or stewardship of NBS but do not engage in economic activity.
- Nature-based products and services may be offered by enterprises or organisations where nature is not a core element of their product/service offering.

#### 4.1.1. Nature-Based Enterprises

Of the total validated survey respondents (148), 108 were SMEs: 76% micro, 22% small, and 2% medium enterprises. To compare, the average enterprise in the EU employs no more than six people, and in 2018, 93% of SMEs were micro [22,54]. An average NBE is thus a larger employer than average EU SMEs, suggesting their work to be more labour intensive. When self-assessing its purpose, most enterprises indicated to be either for profit (40%) or hybrid (44%) as opposed to non-profit (16%). On average, an NBE is owned 61% by private, 28% by third sector organisations, and 11% by public organisations. NBEs

use nature as a core element of their product/service offering for the planning, delivery or stewardship of NBS. With regard to the types of NBS as defined by Eggermont et al. (2015), 51% of NBE operate in type 1 (better use of natural/ protected ecosystems by no or minimal intervention), 61% in type 2 (effective management towards the sustainability and multifunctionality of ecosystems and landscapes), 77% in type 3 (managing ecosystems in intrusive ways or even creating new ecosystems). In the case of the direct use of nature and ecosystem services, nature is used as a resource or input (49%), a supporting service (42%), a cultural or recreational service (38%) or a regulating service (34%).

Several enterprises related to nature-based activities were identified in the literature review, mainly in forestry and tourism. Community benefit enterprises specifically involve communities in governance and management of forests, to provide direct and indirect benefits for the public and the community. Additional objectives include conservation, poverty alleviation, development, cultural revitalisation and political empowerment [55–57]. For-profit forest enterprises can contribute to the conservation and sustainable use of forests while improving the livelihoods of local populations [58]. Traditional forestry companies in Switzerland were found to be shifting their activities from producing ecosystem provisioning services (e.g., timber production) to cultural and supporting services, including recreation and tourism activities and biodiversity conservation practices [59].

Nature-based tourism enterprises offer services in the wilderness or related to wilderness [60]. In the Nordics, activities take place in forests, mountain areas, rivers and lakes and include a large range of services, for example, accommodation, adventure activities, fishing, hunting, educational courses and activities related to food [60,61]. On average, this type of enterprise offers five different services, and are typically rural microenterprises with labour-intensive activities based on local environmental knowledge [62]. Agritourism enterprises are a different type of tourism business that offer farm accommodation and activities. This type of tourism provides authentic or staged interactions between visitors and agriculture on either working or non-working farms [63,64]. As activities of forestry and tourism enterprises tend to be local, there are economic benefits for the local business environment and rural development [58,61,63,65]. Moreover, commercial eco-tourism in protected areas could be beneficial for nature conservation as it provides economic justification for local benefits created [65].

Pro-biodiversity businesses contribute to biodiversity and sustainable use of ecosystems services while being financially profitable. The underlying principle is payments for ecosystem services, where beneficiaries are charged for its use through financial mechanisms, and profit is invested in conservation. Examples of this are wetland and habitat banks [66]. There were also examples found of business models and commercial applications developed by enterprises that use nature sustainably or contribute to NBS. Green care links aspects of the traditional healthcare systems to commercial sectors such as agriculture (e.g., care farming, healing gardens, animal keeping or husbandry) and landscape or nature conservation (e.g., ecotherapy, wilderness therapy). These interventions use both biotic and abiotic elements of nature for well-being and create new value through linking sectors that were not before [67]. Furthermore, commercial applications using data and digital technologies are increasingly used as tools to improve the delivery of benefits provided by nature, in both urban and natural settings, and to enable stakeholder and citizen engagement [65,68,69].

### 4.1.2. Nature-Based Organisations

The 29 nature-based organisations identified in the survey were either public or third sector organisations, 26 are non-profit, and 3 are hybrid. They consist of associations for different environmental causes, education and research organisations, public service companies and government bodies, and 24% are either medium or large organisations, respectively having >50 or >250 employees. On average, a nature-based organisation is owned 48% by third sector, 46% by public, and 6% by private organisations. For 62% of organisations nature is both a direct input, i.e., used by growing, harnessing, harvesting

or sustainably restoring natural ecosystems, and indirect input, i.e., nature contributes to the planning, delivery or stewardship of nature-based solution. Activities include environmental research and education, planning, implementing and management of public green and blue spaces and community gardens, and network and support organisations for NBS.

Examples of nature-based organisations from the literature review include public-private companies, community groups and network organisations in forestry, community gardens and tourism. State-owned forestry enterprises in Europe are often for-profit companies that have sustainable forest management and sustainable wood production as a core activity [70]. In Nordic countries, there is an increased importance of forest environmental services with new business activities being developed, such as renewable energy, real estate and recreation activities [71]. In China, these types of enterprises share similarities with non-profits and focus on societal benefits, such as enhancing ecosystem functions and economic benefits for the community [72]. Urban community gardens comprise both allotment-style and collectively operated gardens that present a wide variety of governance practices and structures as they are often partly governed and funded by the public sector [73]. Support networks for enterprises were found to be important for building connections between entrepreneurs and resources, by creating market opportunities and by advocating for the value of non-market ecological outcomes of these activities [74]. Networks also contributed to streamlining economic data in forestry [75]. In the case of enterprises in nature-based tourism, collective action of businesses has resulted in a more sustainable use of natural resources with less environmental externalities and overexploitation [76].

### 4.1.3. Nature-Based Products and Services

The 11 organisations identified to offer nature-based products and services as part of their activities are private sector and for-profit companies. They consist mainly of engineering and renewable energy companies, and 27% are either medium or large organisations, respectively having >50 or >250 employees. For 73%, nature is primarily an indirect input, and for 27%, it is both direct and indirect. Examples of nature-based products from non-nature-based organisations were found in forestry and finance. Forestry enterprises are privately-owned organisations engaged in the development and utilisation of forest resources for timber production. As part of their management, they might contribute to conservation [77]. In the field of ecological restoration mainly large established companies that partly engaged in these types of activities, as well as new companies, were found. Their activities include aquatic and riparian restoration (18%), wetland restoration (13%), clean-ups and contamination management (13%), terrestrial habitat restoration and management (12%) and mitigation banking (12%), and includes activities on planning, design, and engineering, physical restoration, consulting and monitoring [26]. Moreover, investment activities from multinationals and financial institutions (e.g., banks and pension or investment funds) in biodiversity and conservation are indirectly contributing to NBS delivery, even though they are mainly driven by regulation [78].

### 4.2. Economic Activities of Organisations Delivering NBS

Table 3 provides an overview of the economic activities offered by organisations delivering NBS, and their closest NACE classification. These activities are either directly linked to the provision of ecosystem services, or indirectly by supporting these services, as defined in Section 2.2. More specifically, 56% of NBE use nature directly, with related activities in 7 main categories and 30 sub-categories, while 76% of NBE use nature indirectly, with related activities in 4 main categories and 15 sub-categories. More than 42% of enterprises had activities in more than one category and usually engage in both direct and indirect activities. They offer different activities related to NBS implementation, namely planning and design (78%), advisory (76%), implementation (69%), stewardship (47%),

monitoring (45%) and training (53%). NBE mainly operate nationally (62%), and in urban (83%), peri-urban (67%) and rural (58%) settings.

**Table 3.** Economic activities of organisations delivering nature-based solutions (NBS).

| Economic Activity | No. | Sub-Categories | NBS Type | SME Status | Organisation Type | Closest NACE Classification |
|---|---|---|---|---|---|---|
| *Direct use of nature* | | | | | | |
| Ecosystem creation, restoration and management | 11 | Ecological & landscape restoration<br>Ecosystem conservation and management<br>Biodiversity conservation<br>Reforestation<br>Marine and freshwater ecosystem restoration<br>Marine and freshwater ecosystem conservation and management | Type 1: 8<br>Type 2: 9<br>Type 3: 10 | Micro: 5<br>Small: 6<br>Medium: - | For-profit: 3<br>Hybrid: 5<br>Non-profit: 3 | R. 91.04 Arts, entertainment and recreation: Botanical and zoological gardens and nature reserves activities |
| NBS for green buildings | 10 | Living green roofs and facades<br>Living green wall indoor<br>Living green walls outdoor | Type 1: 3<br>Type 2: 3<br>Type 3: 9 | Micro: 6<br>Small: 4<br>Medium: - | For-profit: 5<br>Hybrid: 5<br>Non-profit: - | N. 81.3 Administrative and support service activities: Landscape service activities |
| NBS for public and urban spaces | 28 | Green areas, parks and gardens<br>Green infrastructure<br>Green space management<br>Urban forestry<br>Urban regeneration projects | Type 1: 13<br>Type 2: 17<br>Type 3: 23 | Micro: 23<br>Small: 5<br>Medium: - | For-profit: 11<br>Hybrid: 11<br>Non-profit: 6 | N. 81.3 Administrative and support service activities: Landscape service activities |
| NBS for water management and treatment | 14 | Natural flood & surface water management<br>Urban green and blue infrastructure<br>Urban water management<br>Wastewater management | Type 1: 6<br>Type 2: 7<br>Type 3: 12 | Micro: 10<br>Small: 4<br>Medium: - | For-profit: 6<br>Hybrid: 6<br>Non-profit: 2 | E. 36 Water supply; sewerage, waste management and remediation activities: Water collection, treatment and supply |
| Sustainable agriculture & food production | 15 | Agroforestry<br>Beekeeping<br>Horticulture<br>Plant and soil improvement<br>Regenerative farming | Type 1: 8<br>Type 2: 10<br>Type 3: 10 | Micro: 13<br>Small: 2<br>Medium: - | For-profit: 3<br>Hybrid: 10<br>Non-profit: 2 | A.1 and 1.3 Agriculture, forestry and fishing: Crop and animal production, hunting and related service activities; Plant propagation |
| Sustainable forestry and biomaterials | 7 | Sustainable forestry<br>Biomaterials for construction<br>Biomaterials for food preservation | Type 1: 5<br>Type 2: 3<br>Type 3: 6 | Micro: 4<br>Small: 3<br>Medium: - | For-profit: 3<br>Hybrid: 3<br>Non-profit:1 | A. 2.1 Agriculture, forestry and fishing: Silviculture and other forestry activities<br>C. 16 Manufacturing: Manufacturing of wood and of products of wood and cork |
| Sustainable tourism and health & wellbeing | 3 | NBS for health & wellbeing<br>Agritourism<br>Eco-tourism and nature-based tourism<br>Forestry tourism | Type 1: 3<br>Type 2: 1<br>Type 3: 1 | Micro: 3<br>Small: -<br>Medium: - | For-profit: -<br>Hybrid: 3<br>Non-profit: - | N.79 Administrative and support service activities: Travel agency, tour operator and other reservation service and related activities<br>Q. 86.9 Human health and social work activities: Other human health activities |
| *Indirect use of nature* | | | | | | |
| Advisory services | 41 | Biodiversity and ecosystems<br>Urban greening design & planning<br>Landscape architecture<br>Water management<br>Community engagement for NBS | Type 1: 21<br>Type 2: 28<br>Type 3: 33 | Micro: 32<br>Small: 8<br>Medium: 1 | For-profit:16<br>Hybrid: 16<br>Non-profit: 9 | M. 71 and 74 Professional, scientific and technical activities: Architectural and engineering activities, technical testing and analysis; Other professional, scientific and technical activities |
| Education, research & innovation activities | 18 | Ecological research<br>Environmental awareness education<br>Research & innovation projects<br>Vocational & skills training | Type 1: 8<br>Type 2: 8<br>Type 3: 11 | Micro: 16<br>Small: 2<br>Medium: - | For-profit: 4<br>Hybrid: 9<br>Non-profit:5 | M. 72 Professional, scientific and technical activities: Research and experimental development on natural sciences and engineering<br>P. 85 Education: Other education |
| Financial services | 3 | Carbon offsetting<br>Investment for biodiversity and conservation<br>Natural capital accounting | Type 1: 2<br>Type 2: 3<br>Type 3: 3 | Micro: 2<br>Small: 1<br>Medium: - | For-profit: 1<br>Hybrid: 2<br>Non-profit: - | K. 64.9 Financial and insurance activities: Other financial service activities |
| Smart technology, monitoring and assessment of NBS | 20 | Smart technology solutions for NBS<br>Environmental monitoring<br>Spatial tools for environment | Type 1: 10<br>Type 2: 17<br>Type 3: 11 | Micro: 15<br>Small: 4<br>Medium: 1 | For-profit: 13<br>Hybrid: 4<br>Non-profit: 3 | J. 62 Information and communication: Computer programming, consultancy and related activities |

### 4.2.1. Categories of Direct Nature-Based Activities

Most enterprises using or enhancing nature directly for or as a result of their products or services are micro or small enterprises, generally have a commercial focus and are for-profit or hybrid. Activities under ecosystem creation, restoration and management focus on all NBS types, not only natural ecosystems (NBS type 1), but also urban ecosystems, such as allotments, community gardens and derelict areas. Some of the activities thus overlap with NBS for public and urban spaces which include urban regeneration projects in addition to green areas, parks, gardens and playgrounds, green infrastructure, and urban forestry. NBS for green buildings mainly relate to NBS for air purification and water retention, and enterprises are involved in different activities around the design, implementation and maintenance of their products. NBS for water management and treatment includes natural solutions for the management of flood and surface water, in rural, peri-urban and urban contexts, and wastewater management and treatment, and resource recovery. Non-profit enterprises in ecosystem creation, restoration and management mainly focus on nature conservation and protection, and in NBS for public and urban spaces on greening areas as a benefit for society. NBEs for public and urban spaces combine activities with green infrastructure, monitoring and smart technology activities. The NACE classifications for the activities identified in the three categories above do not accurately represent the nature of these organisations core objectives and activities as these activities are categorised respectively under the categories 'Arts, entertainment and recreation' and 'Administrative and support service activities'.

For the categories of sustainable agriculture and food production and sustainable forestry and biomaterials, the NACE classification - while more appropriate - still lacks the important distinction from the sustainability perspective. Activities under the first category of sustainable agriculture and food production include sustainable food production activities that contribute to ecosystem services and biodiversity and include agroforestry, regenerative agriculture and horticulture, beekeeping and natural plant and soil improvement. Activities included under Sustainable forestry and biomaterials use nature as a sustainable input for construction and manufacturing for buildings, industry and products. Examples are the manufacturing and application of biomaterials for construction of agricultural and irrigation systems (such as hydroponics), growing algae for food products, and sustainable forestry. Finally, sustainable tourism and health and well-being include the use of nature for leisure and well-being activities, including eco-tourism activities and outdoor workshops for wellbeing purposes, such as forest bathing. Nature is used directly for these purposes. The NACE classification does include such activities, classified respectively as 'Administrative and support service activities' and 'Other human health activities'. Again, this classification seems to miss the nature-based and sustainability focus at the heart of NBE activities.

### 4.2.2. Categories of Indirect Nature-Based Activities

Similar to the enterprises directly using nature, most enterprises in this category are micro or small enterprises and are either for-profit or hybrid, with their turnover mostly derived from the private and public sector. Advisory services and education, research and innovation activities are broad categories of activity, and the closest NACE categories do seem to encompass them broadly. Activities in the latter focus on knowledge collection and dissemination, and, among others, include innovation and feasibility projects for NBS from environmental and social perspectives. Examples under advisory services include technical activities in the planning, design, implementation and management of NBS, as well as social components of this, for example, community engagement. The non-profit enterprises found in these two categories are primarily focused on activities with social components, such as the creation of environmental awareness and activities related to the inclusion and mobilisation of individuals or communities. An example of a financial service enterprise is an enterprise offering subscriptions to businesses and individuals to finance ecosystem restoration projects as a way to offset carbon impact, mainly in the form

of reforestation. Activities under smart technology, monitoring and assessment of NBS use satellite imagery, environmental sensors, spatial tools and data analytics, e.g., for creating an inventory of tree species or analysis tree and soil health. These types of applications are applied in all three types of NBS, with the highest representation in type 2, semi-natural ecosystems. In an urban context, smart solutions are combined with small green areas such as green roofs, and data analytics is used to provide insights for increasing effectiveness. The NACE classifications for these activities and financial services do not account for the level of detail of these activities, nor acknowledge the business models used.

## 5. Discussion

### 5.1. Research Findings

The typology for organisations delivering nature-based solutions is based on the underlying concepts of enterprise, i.e., their engagement in economic activity, and on nature-based solutions, i.e., the use of nature as a core activity. The latter is difficult to quantify and evaluate, and moreover, relied mostly on self-reporting of the survey respondents. The underlying concepts are derived from our definition of nature-based enterprise (Section 1), and one of the aims of this research was to test its robustness. Although we are confident that this definition has provided a useful foundation for this study, more responses and engagement from academics and practitioners is required to further test its validity.

The typology as a result of the research gives insight into the different types of organisations contributing to the implementation of NBS and shows the importance of the private, public and third sector. Data collected in this research were, however, mainly focused on the private sector. We recognise that nature-based organisations-though not active on economic markets-play an important role in the financing of, and in providing space and regulatory frameworks for NBS (e.g., city departments), as well as for the financing of nature, mainly for nature conservation, and for the empowerment of local communities (e.g., NGO environmental charities).

The categories of economic activities identified in this research are inadequately covered by NACE, while this is essential for the economic assessment, financial investment and market development of sectors. The categorisation of activities proposed in this research is a first step towards a classification. However, as the study was conducted within the context of the Connecting Nature project, that focuses on urban NBS, certain relevant activities have not been captured. The focus has primarily been on 'green' terrestrial solutions as opposed to 'blue' solutions, and activities such as sustainable aquaculture practices (i.e., seaweed farming) were not found in the research.

### 5.2. Limitations

As in all studies, the limitations of this study have to be acknowledged. First, the inclusion and exclusion criteria used in the systematic literature review can be criticised. Due to the use of the term nature-based solutions in literature, papers from before 2010 were excluded. Expert recommendations and hand-searching resulted in studies on nature entrepreneurship in forestry and tourism from to be included as additional literature. Although an extensive list of relevant keywords was used in the search, we cannot guarantee that all work relevant to nature-based enterprise is included, as different terminology may have been used. Second, the survey relied on self-assessment, and its representativeness can be criticised. This self-reporting may have resulted in companies reporting to engage in 'greenwashing' and overstate their nature-related activities. The survey has a relatively small sample size, and its geographic distribution is mainly focused on Europe. Therefore, the economic activities found in this research might not provide a complete overview. Third, concerns could be raised regarding the objectivity of the data analysis as literature reviews, empirical data collection, analysis, and interpretation are subjective. Other groups of researchers might interpret the results differently or come up with other dimensions for

the typology or categorisation of economic activity. This limitation has been addressed through cross-validation with both an expert and practitioner panel.

Despite the study's limitations, we believe that the literature included in the review represents an accurate sample of the current themes in academic research and that the empirical data collected in the survey present a broad overview of relevant categories of activities by organisations delivering NBS, and that the typology and categorisation represent a useful building block to advance the understanding of nature-based enterprise.

*5.3. Future Research*

To advance theory surrounding the organisational typology of organisations delivery NBS and the categorisation of economic activities proposed in this paper, further empirical studies would be beneficial. In particular larger empirical studies, studies outside of Europe, and more detailed studies of the individual organisation types identified in this research. Furthermore, research exploring the economic value created by nature-based enterprises is important for the assessment and market development of NBE, as these are currently not included in existing industrial classifications. Moreover, the recognition of economic value creation could contribute to the urgently needed alternative valuation of nature and ecosystem services.

The contribution of nature-based enterprises to job creation is an important dimension of economic value creation which warrants further exploration. While all survey respondents in this study fell within the EC definition of SME, this contrasts with other studies (e.g., [26]) that suggest larger employers are involved in this field as well. Moreover, at the SME level, nature-based enterprises were found to have more employees than the EU average. In terms of future employment policy, more in-depth insights into organisation characteristics are important; however, further empirical studies are needed to validate these findings. Furthermore, research efforts directed at the environmental and social value created by nature-based enterprises are vital in realising their potential to contribute to global societal challenges, and objectives of, for example, the EU Green Deal and the UN Sustainable Development Goals.

Finally, little is known about the start-up and growth trajectories of nature-based enterprises and how this compares with that of other industry sectors. Research in this direction would benefit from more in-depth qualitative analysis revealing deeper insights about aspects such as entrepreneurial motivation, barriers and enablers to growth. Some data on these aspects were collected as part of this wider study, was out of the scope of this paper.

## 6. Conclusions

In this study we have proposed a typology for organisations involved in the planning, delivery, and stewardship of nature-based solutions, based on economic activity and the sustainable use of nature. It introduces nature-based enterprise—defined as enterprises that use nature as a core activity in their product/service offering and that engage in economic activity—nature-based organisations (core use of nature, but no economic activity) and nature-based products and services delivered by enterprises or organisations, where nature is not at the core of activities. In addition, it was identified that these organisation types were involved in 11 categories of economic activities, 7 in which nature is used directly, and 4 where it is used indirectly. Comparing our typology of nature-based economic activities with NACE, the European classification for industry activities, we found that the latter did not contribute to the understanding of these activities, and inadequately addressed the essential aspect of environmental sustainability.

The typology of nature-based enterprises provides the groundwork for further studies quantifying the economic, social and environmental contribution of this type of organisation. Moreover, the recognition of nature-based enterprises as important actors in the implementation of nature-based solutions is an essential first step in market creation for the products and services they offer. Although more research is needed for the identification of

barriers and enablers in the start-up and growth of nature-based enterprise, the findings in this research are crucial for policy development for private sector involvement for wider NBS adoption.

To summarise, this study of the characteristics of organisations delivering NBS and the resulting typology and categorisation of economic activities, suggests that this sector should be considered in future policy as a stand-alone sector with significant potential to contribute to the EU goal of achieving a climate-neutral economy by 2050.

**Author Contributions:** Conceptualisation, E.D.K., S.M., M.-L.R. and M.J.C.; Formal analysis, E.D.K. and S.M.; Investigation, E.D.K.; Methodology, E.D.K. and S.M.; Writing—original draft, E.D.K.; Writing—review & editing, S.M., M.-L.R., M.J.C. and F.P. All authors have read and agreed to the published version of the manuscript.

**Funding:** This research is conducted as part of the Connecting Nature project and is funded by the Horizon 2020 Framework Programme of the European Union under Grant Agreement number 730222.

**Institutional Review Board Statement:** Not applicable.

**Informed Consent Statement:** Not applicable.

**Data Availability Statement:** The data presented in this study is available upon request from the corresponding author.

**Conflicts of Interest:** The authors declare no conflict of interest. The funders had no role in the design of the study; in the collection, analyses, or interpretation of data; in the writing of the manuscript, or in the decision to publish the results.

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
