# Peer review of "Innovating with Nature: From Nature-Based Solutions to Nature-Based Enterprises"

_sustainability, doi:10.3390/su13031263_

Round 1

Reviewer 1 Report

The research is devoted to very actual aspect of sustainable development: innovating with Nature: from Nature-Based Solutions to Nature-Based Enterprises. That’s all world problem, because many scientists and politicians point the problem of global climate but nature-based enterprises are out of their focus, especially forestry and wood-working, which are involved in the process of deposing CO2 but they are not in green economy and green policy focus in many countries, even developed one.

The manuscript structured logically due to principle of IMRAD. All part of the article connected logically and structured good.

Methodology is described enough detailed and presented good. The only part which should be improved is connected with respondents characteristic: what is the size of interviewed enterprises, which location etc.

In the part of analyses the survey was done only with 173 enterprise respondents. For the level of Q2 ( Sustainability) it should be improved till 300 and in some article the research collected data from 1 000 respondents. So authors should think and involved more respondents for better quality of the research results.

In literacy review it should be added actual information about Green Deal (signed on 9 of December 2020) and how it will influence on the research topic.

Finally, I would like to summarize that the manuscript is good and after bellowed small changes it could be appropriate for Sustainabilty.

Reviewer 2 Report

The authors have done a good job in studying the topic and compiling the manuscript. Find below are my recommendations for improving this manuscript.

Introduction and theoretical background 

The goal of the study is explicitly stated and supported with a clear rationale. The research questions are appropriately developed and aligned with the primary goal of study. The significance of the research study is well established, and the overall theoretical research framework relates well with the study context.  

Literature is thoroughly presented and discussed to set the context of the research problem. Diverse literature is used to present contributing as well as competing perspectives.  

It is suggested that authors add examples of ‘Ecological and environmental enterprises’ (see lines 118-120) as done for Eco-enterprises and green enterprises. This will help authors to stay consistent in their writing. 

Methodology 

The author used an appropriate research design. Data collection and analysis techniques are accurately used.  

It is recommended that authors include a description of Data Analysis (3.3)especially sub-sections 3.3.1 and 3.3.2. It is confusing to follow numbers presented in Figure 1 under data Analysis-Thematic analysis of the activities section. Discussion on economic activities (n=176) and 1st, 2nd, and 3rd order categorization under section 3.3.1 and 3.3.2 will bring more clarity on research methods. 

It is recommended to use the terms structured questions instead of ‘pre-determined answers’, and ‘open-ended questions’ instead of ‘open questions (see lines 220). 

Research Findings/Results  

Overall, the author has done well in presenting results. There is a logical flow in the result presentation.  

Under section 4.1.1, “all 108 nature-based enterprise...” (see line 334) sounds confusing. It need to be rephrased to bring clarity in writing such as of the total (148) nature-based enterprises selected for the study, 108 enterprises were SMEs 

Discussion  

The discussion was meaningful and logical. Authors interpreted the findings well and it does contribute to the existing body of knowledge. Authors did well in identifying problems with the current NACE classification. 

Research contribution, limitation and future directions are clearly stated.
